# scTIGER: A Deep-Learning Method for Inferring Gene Regulatory Networks from Case versus Control scRNA-seq Datasets

**DOI:** 10.3390/ijms241713339

**Published:** 2023-08-28

**Authors:** Madison Dautle, Shaoqiang Zhang, Yong Chen

**Affiliations:** 1Department of Biological and Biomedical Sciences, Rowan University, Glassboro, NJ 08028, USA; dautle14@students.rowan.edu; 2College of Computer and Information Engineering, Tianjin Normal University, Tianjin 300387, China

**Keywords:** scRNA-seq, gene regulatory network, deep learning, gene co-differential expression network, memory formation, prostate cancer

## Abstract

Inferring gene regulatory networks (GRNs) from single-cell RNA-seq (scRNA-seq) data is an important computational question to find regulatory mechanisms involved in fundamental cellular processes. Although many computational methods have been designed to predict GRNs from scRNA-seq data, they usually have high false positive rates and none infer GRNs by directly using the paired datasets of case-versus-control experiments. Here we present a novel deep-learning-based method, named scTIGER, for GRN detection by using the co-differential relationships of gene expression profiles in paired scRNA-seq datasets. scTIGER employs cell-type-based pseudotiming, an attention-based convolutional neural network method and permutation-based significance testing for inferring GRNs among gene modules. As state-of-the-art applications, we first applied scTIGER to scRNA-seq datasets of prostate cancer cells, and successfully identified the dynamic regulatory networks of AR, ERG, PTEN and ATF3 for same-cell type between prostatic cancerous and normal conditions, and two-cell types within the prostatic cancerous environment. We then applied scTIGER to scRNA-seq data from neurons with and without fear memory and detected specific regulatory networks for BDNF, CREB1 and MAPK4. Additionally, scTIGER demonstrates robustness against high levels of dropout noise in scRNA-seq data.

## 1. Introduction

Novel single-cell RNA (scRNA) technologies have enabled the monitoring of biological systems and complex tissues to deeply analyze characterizing rare cell populations, examining molecular heterogeneity across cell populations, and investigating the transcriptional dynamics throughout disease pathology [1,2,3,4]. Among diverse computational tasks in scRNA-seq data analysis, inferring gene regulatory networks (GRNs) remains as one of the major challenges [5,6,7]. GRNs refer to the complex systems of interactions among genes and other regulatory molecules that control the expression of genes in cells or organisms. In network terminology, GRNs are constructed by nodes representing regulators and their target genes, and edges representing the regulatory relationships between genes. GRNs are important for the proper functioning and development of organisms and play a critical role in controlling many biological processes, including cell differentiation, tissue development, and response to environmental stimuli [8,9]. By understanding how GRNs work, researchers can gain insights into the underlying mechanisms of disease, as well as develop new approaches for treating or preventing various conditions [10,11]. Compared with bulk RNA-seq, inferring GRNs from scRNA-seq data has two major advantages. First, scRNA-seq allows for the capture of gene expression patterns at the single-cell level, facilitating the examination of dynamic changes in gene regulatory networks during development or in response to perturbations, which in turn provides a more comprehensive understanding of cellular behavior. Second, scRNA-seq enables the identification of different cell types and subpopulations within tissues, allowing for a more accurate representation of gene regulatory networks [2,3,5,6,12].

Although inferring GRNs from scRNA-seq data provides a promising way for understanding the complex regulatory mechanisms that control gene expression and cellular behavior, it presents computational challenges in designing network inference algorithms that can handle high-dimensional, noisy data and identify causal relationships between genes [5,6,7]. To address this challenge, many computational approaches have been specifically developed for inferring networks from scRNA-seq data. For instance, SCENIC [13] combines gene expression data with information from gene regulatory databases to identify transcription factors that drive the expression of downstream target genes. SCODE [14] uses a deconvolution approach to identify the expression levels of transcription factors and their target genes within each cell and integrates these expression levels with prior knowledge of known regulatory interactions to infer a GRN. SCOUP [15] employs a combination of unsupervised clustering and network inference methods to identify cell subpopulations and infer GRNs. PIDC [16] uses the partial information decomposition method for inferring dynamic GRNs from scRNA-seq time-series data. GENIE3 [17] is a machine learning method that uses a random forest algorithm to identify the most important regulatory interactions between genes. WGCNA [18] constructs a co-expression network by calculating the correlation between gene expression profiles and identifies modules of co-expressed genes. It then infers GRNs within each module by calculating the partial correlation between genes. ShareNet [19] employs a Bayesian framework for boosting the accuracy of cell-type-specific gene regulatory networks for a given single-cell dataset. To improve the performance of GRN inference, new methods integrated additional single-cell omics data to filter and/or highlight true regulatory relationships among genes [20]. For example, NetAct [21] constructs core transcription factor regulatory networks using both transcriptomics data and literature-based transcription factor-target databases. DeepDRIM [22] is a novel supervised method based on deep neural networks to utilize the image of the target TF-gene pair and the ones of the potential neighbors to reconstruct GRN from scRNA-seq data. iGRN [23] incorporates multi-omics data (gene expressions, copy number variations, DNA methylations, and their genetic interactions) to construct gene regulatory networks.

While there has been significant progress in inferring GRN from scRNA-seq using computational methods, there are still several limitations that need to be addressed. First, there is a lack of methods for inferring condition-specific GRNs from case-versus-control scRNA-seq experiments (e.g., healthy vs. cancer, or perturbation vs. non-perturbation) that are widely applied in biological research. Such analysis is important not only for identifying dysregulated regulatory networks in disease states, but also for investigating molecular mechanisms in biological systems with environmental perturbations. Second, the accuracy of GRN inference algorithms is mainly affected by high rates of false-positive and false-negative interactions [7]. False-positive interactions are correlations that are detected as significant but are actually due to chance or confounding factors, while false-negative interactions are true correlations that are not detected as significant due to noise or other limitations. The high false-positive rates are observed in many correlation-based methods such as WGCNA [18], SCENIC [13] and SCODE [14], which mainly rely on pairwise correlation measures such as Pearson correlation, Spearman correlation, or mutual information to identify co-expression patterns between genes. Although the correlation matrix of gene expression levels can be used to construct a network or graph representing regulatory interactions between genes, correlation does not always imply causation, and spurious correlations can arise due to confounding factors or other biases. For example, two genes may be co-expressed during the cell cycle in the same cell type, but one gene may not actually regulate the other. Third, current scRNA-seq-based research lacks time-series data and/or additional omics data, which makes it difficult to identify causal relationships. While GRN inference algorithms can uncover statistical correlations between genes, they do not offer direct evidence of causality. To establish causality in gene expression data, researchers traditionally rely on large time-series datasets, which are not currently available in scRNA-seq experiments. Moreover, many scRNA-seq experiments lack paired omics data such as chromatin accessibility, histone modifications, or genetic variants, which makes certain methods (i.e., iGRN, DeepDRIM, and NetAct) impractical and of lower precision. As a result, the lack of scRNA-seq time-series data and/or additional omics data poses a significant challenge in identifying key regulatory nodes, limits the effectiveness of computational analysis, and reduces the interpretability of inferred networks. Overall, these limitations underscore the need for designing computational methods for GRN inference from abundant case vs. control scRNA-seq datasets, with a focus on improving accuracy and interpretability.

In this study, we propose a novel deep-learning-based method named scTIGER (single-cell temporal inference of gene expression regulation) for inferring GRNs from case versus control scRNA-seq datasets. scTIGER initially constructs the gene co-differential expression network (GCEN), comprising edges only among genes that exhibit coupled changes between the two conditions. scTIGER combines three major strategies: cell-type-based pseudotiming [24], temporal causal discovery framework [25], and permutation-based significance testing models, to infer causal relationships among genes. To benchmark this state-of-the-art method, we first applied scTIGER to scRNA-seq datasets from patient samples with prostate cancer and specifically focused on identifying the dynamic regulatory networks of AR, ERG, PTEN, and ATF3 for the same-cell type between prostatic cancerous and normal conditions, and two-cell types within the cancerous prostatic environment. We then applied scTIGER to scRNA-seq data from neurons with and without fear memory to detect specific regulatory networks of BDNF, CREB1, and MAPK4 that are involved in remote memory formation in the mouse brain. In both applications, scTIGER not only predicted condition-specific GRNs for the same cell types, but also cell-specific GRNs across different cell types within the same condition. These results demonstrate that scTIGER is an effective method for detecting regulatory relationships through general case vs. control scRNA-seq datasets.

## 2. Results

### 2.1. scTIGER Pipeline

The scTIGER method employs two innovative strategies to construct GRNs from paired scRNA-seq datasets, such as case vs. control (Figure 1a). The first strategy addresses the limitation of time-series data by utilizing pseudotiming, which orders the cells and provides a chronological view of the whole-genome gene expression profile. This enables the identification of ordered changes in gene expression. The second strategy involves filtering false-positive regulations by considering the correlations of differential expression vectors within given gene sets. As depicted in Figure 1b, gene pairs lacking high co-differential relationships are discarded, even if they exhibit high correlations of expressions under two conditions. This approach helps mitigate potential false positives that may be predicted by methods solely relying on gene expression correlations. When applied to the same-cell type under two different conditions, scTIGER enables the investigation of GRNs through which the cell modifies its regulatory patterns in response to changes in conditions. Additionally, when applied to two different cell types under the same experimental condition, scTIGER provides insight into how these cell types exhibit similar mechanisms within a specific condition (Figure 1c).

The scTIGER pipeline is designed for ease of use in a Linux environment, allowing for simple single-line command execution. It offers flexibility by accepting various user-defined parameters, including single or multiple genes of interest, the number of permutations, the selection of top genes for attention-based deep learning, the threshold for dropout percentage permitted for gene inclusion in the analysis, significance levels for predicted regulations, and the maximum number of time steps allowed for an inferred edge. Additionally, scTIGER provides the option to utilize CUDA for accelerated processing if desired.

The output of scTIGER also prioritizes a user-friendly experience. Instead of generating pre-graphed and unmodifiable images, scTIGER produces GraphML files that can be imported into the user’s preferred network graphing software. Individual files are generated for each gene of interest, and a merged file consolidates the global interactions from all genes into a cohesive network. scTIGER also provides histograms illustrating the number of interactions detected at specific frequencies, offering visual assistance in determining statistically significant interactions for inclusion in the network file. Furthermore, the scTIGER package provides a Cytoscape-style file for visualizing causal interactions along with their regulation types. The raw output files contain information on causal interactions, regulation types, time delays between regulations, and the frequency of detection. The scTIGER package also includes sample datasets, sample commands for installation testing, and replication of the analyses conducted here, along with sample outputs for comparison.

### 2.2. Comparing scTIGER Performance with Other Methods

To benchmark scTIGER’s performance in detecting GRNs, we compared scTIGER’s performance to five existing models, i.e., PIDC [16], SINCERITIES [26], SCODE [14], GRNBoost2 [27], and PPCOR [28] whose properties are detailed in Table 1. We utilized synthetic and curated datasets that were previously used to compare many GRN models [29], including a linear trajectory (LI), cyclical trajectory (CY), hematopoietic stem cell differentiation (HSC), and mammalian cortical area development (mCAD). Please note, due to the lack of benchmarking datasets for paired case versus control data, the benchmarking only covers the inference of edges using the deep-learning algorithm, permutation, and significance threshold calculation steps.

When benchmarking scTIGER against current models, we calculated the precision, recall, F1 score, and specificity for each on four datasets. We find that scTIGER outperforms the other five existing models in various criteria across four different datasets (Table 2). In terms of recall, which measures the ability to identify true positives, scTIGER consistently achieves high scores in three datasets (LI, CY, and HSC), especially on LI and CY trajectories where it achieves a perfect recall score of 1. scTIGER also achieves the highest precision scores on LI, CY, and HSC with scores of 1, 0.75, and 0.56, respectively. While some methods, like GRNBoost2, perform well on specific datasets (e.g., mCAD), scTIGER maintains its competitive edge by achieving high specificity on the LI and CY datasets. The F1 score, which considers both precision and recall, demonstrates that scTIGER strikes well-balanced performance across various datasets, outperforming other methods on LI, CY, and HSC with F1 scores of 1, 0.857, and 0.609, respectively. Thus, scTIGER consistently performs well across different datasets, making it a robust and reliable choice for inferring gene regulatory networks.

### 2.3. scTIGER Detected Potential AR Regulatory Network in Prostatic Cancerous vs. Normal Cells

To showcase the capability of scTIGER in exploring novel regulations, we utilized scRNA-seq datasets obtained from patient samples with prostate cancer [30]. Our analysis encompassed the identification of regulatory differences between prostate cancer (PCa) samples and their corresponding normal prostate cells. Additionally, we investigated common regulations occurring between tumorigenic endothelial cells (ECs) and smooth muscle cells (SMCs). Notably, ECs play a pivotal role in PCa progression and metastasis, serving as a crucial barrier that cancer cells must breach to access the bloodstream. This transition from non-metastatic to metastatic cancers marks a critical turning point [31,32,33,34,35]. Furthermore, SMCs represent another vital barrier against PCa progression. The prostate is extensively enveloped by smooth muscle, necessitating cancer cells to infiltrate and traverse these cells to reach the endothelial cells. Consequently, understanding gene regulation surrounding PCa metastasis demands insights into the interplay within this cell type as well [36,37]. In this section, our focus centered on three extensively studied cell-specific and/or oncogenes in ECs: AR, ERG, and PTEN. Specifically, we applied scTIGER to investigate the regulatory dynamics of these genes, aiming to uncover their roles and interactions within the context of prostate cancer.

The AR gene encodes the androgen receptor protein, which is a steroid-hormone-activated master regulator in prostate tissues [38,39] and plays a critical role in prostate cancer progression [40]. AR-mediated signaling is a distinctive hallmark of PCa, characterized by hyperactivation of the AR receptor, which leads to increased local androgen synthesis, upregulation of coactivators, and downregulation of corepressors [40,41]. scTIGER analysis identified 41 significant interactions directly linked to AR (Figure 2a). Several of the detected interactions, such as NOB1, REV1, and ROBO1, showed upregulation by AR, while CAV2 and BAG3 exhibited downregulation by AR. Over 85% of these interactions have been reported in the literature (Figure 2a, Appendix A). Further investigation was conducted to understand the functions of these AR-regulated genes. First, AR was found to regulate AHNAK2, CNRIP1, GPM6A, and CHP1, which are associated with calcium regulation, as well as ARF5, GCC2, and UBL3, which play a role in vesicle trafficking (Figure 2a). These findings align with previous studies indicating that alterations in calcium regulation and vesicle trafficking can promote the growth and metastasis of PCa [42,43,44]. Second, scTIGER detected AR-regulated genes associated with changes in the initiation steps of translation, resulting in an overall decrease in mRNA translation in PCa [45,46]. Specifically, scTIGER identified AR downregulating NSUN6 expression in PCa, and it has been shown that mRNAs associated with NSUN6 exhibit higher rates of translation, thus causing a decrease in overall translation [47]. The observed increase in AR expression in PCa would consequently lead to a decrease in NSUN6 expression and translation rates. Additionally, upregulation of AR in PCa is co-regulated by YTHDF3 [48]. scTIGER identified AR as upregulating YTHDF3 expression, suggesting a potential indirect positive feedback loop between the two genes, promoting the upregulation of both AR and YTHDF3 to create an environment conducive to PCa (Figure 2a). Furthermore, NOB1, a gene associated with ribosome formation, is a known oncogene that is upregulated in PCa and associated with cell proliferation [49]. scTIGER revealed the upregulation of NOB1 expression by AR in PCa, aligning with literature findings on PCa progression. Third, scTIGER identified the regulation of DOCK4 by AR. DOCK4 is a key gene involved in PCa metastasis through cytoskeletal regulatory adjustments [50,51,52]. ChIP-seq experiments have shown that AR targets DOCK4, and previous research has demonstrated that DOCK4 is overexpressed in PCa, leading to metastatic prostate cancer. Reducing the concentration of DOCK4 to normal levels has been shown to decrease cell invasion [53,54]. Here, scTIGER confirmed the regulation of DOCK4 by AR (Figure 2a).

Previous studies have demonstrated that the AR signaling pathway is associated with decreased PTEN expression [41]. PTEN encodes an enzyme that acts as a tumor suppressor by inhibiting the P13K/AKT signaling pathway, which is known to promote PCa survival and growth [41,55]. In our analysis, we focused on investigating PTEN using scTIGER and identified several significant interactions directly related to PTEN (Figure 2b). Notably, some of these interactions include the upregulation of RAB4A, LAMA5, and STX12 by PTEN, and the downregulation of ACAP2 and MGLL (Figure 2b). Out of the 23 interactions detected with a direct link to PTEN, 11 interactions were supported by the literature (Figure 2b, Appendix A). One noteworthy downstream regulatory element of PTEN is GOLGA2, which requires PTEN to prevent the activation of an alternative splicing site for proper function. Activation of this alternative splicing site disrupts the Golgi secretory pathway, contributing to PCa metastasis [56]. scTIGER successfully detected the positive regulatory relationship between PTEN and GOLGA2 (Figure 2b). Another important gene is SLC39A7, a zinc regulator. Accumulation of zinc is crucial for normal prostate maintenance, but it is often found to be underexpressed in PCa [57,58,59]. scTIGER identified a positive regulatory relationship between PTEN and SLC39A7, suggesting that the downregulation of SLC39A7 could be linked to decreased expression of PTEN (Figure 2b).

For ERG, scTIGER detected a few significant interactions directly related to this gene (Figure 2c). Among the four genes with an edge to ERG, two interactions were supported by the literature (Appendix A). All these interactions were targets of ERG and exhibited downregulation patterns. The supported interactions involved ERG downregulating HCLS1 and DDRGK1 (Figure 2c, Appendix A).

To explore potential functional connections among AR, PTEN, and ERG, we examined the combined interactions of their predicted GRNs. Even when considering only genes within one step of the genes of interest, we observed the regulation of SH3BGRL2 by both PTEN and AR (Appendix A). In both cases, this regulation would lead to a decrease in the gene product. Increased AR expression in PCa results in the downregulation of SH3BGRL2 through MMP24OS, while decreased PTEN levels in PCa correspond to the downregulation of SH3BGRL2 (Appendix A). This concept finds partial support in previous studies indicating that SH3BGRL2 acts as a tumor suppressor in clear cell renal cell carcinoma [60]. Given the potential tumor suppressor functionality of SH3BGRL2 in PCa, the dual downregulation observed here presents a plausible hypothesis. These findings highlight that scTIGER not only recovers known edges and uncovers new ones but also identifies potentially important regulatory patterns by combining the results from multiple genes of interest.

### 2.4. scTIGER Detected Potential AR Regulatory Network in Different Cell Types in Prostatic Environments

We then utilized scTIGER to identify the regulations by using the co-differential expression of ECs and SMCs in PCa (Figure 3). Understanding such regulations between cell types in a disease is essential because cell-specific gene co-differential changes provide insights into genes that are uniformly expressed across different cancer cell types. It has the potential to facilitate the tracking of dedifferentiation or transdifferentiation processes in PCa or other diseases [61,62,63]. A total of 51 significant interactions were identified for AR, among which more than 65% have been documented in the literature (Figure 3a, Appendix A). The interactions detected by scTIGER reveal broad functional regulations in ECs and SMCs corresponding to tumor suppression and apoptosis, transcription and translation, post-translational modifications, DNA repair and replication, cell adhesion, and cellular transport.

When examining the interactions associated with PTEN in ECs and SMCs in PCa, we detected 9 significant regulations, and 5 of them were documented in the literature (Figure 3b, Appendix A). These interactions involve the immune response, cell surface receptor signaling, mitochondrial function, cell adhesion and differentiation, post-transcriptional modifications, and Golgi surface recycling. For ATF3, scTIGER detected 23 interactions and we found that 60% of the significantly detected interactions were verified in the literature (Figure 3c, Appendix A). These ATF3-regulated genes are enriched in tumor suppression and apoptosis, transcription and translation, cellular transport, mitochondrial function, and protein degradation. When merging the three GRNs together, a connected network was obtained (Appendix A). The merged network reveals that multiple genes (PDCD7, CFI, KIF1B) are commonly regulated by both AR and ATF3, and a gene (GLRX5) is commonly regulated by both ATF3 and PTEN.

### 2.5. scTIGER Detected Potential Regulatory Networks in Neurons with and without Memories

We further employed scTIGER on a dataset containing neural and neural-associated cells from fear-conditioned mice, with non-conditioned mice serving as the control. Previously, we identified a set of differentially expressed genes (DEGs) that may be involved in remote memory formation [64], but the exploration of regulatory relationships among these DEGs has been limited. In this study, we focused on BDNF, CREB1, and MAPK4, which have documented impacts on remote memory formation, consolidation, recall, or reconsolidation. BDNF is responsible for regulating inhibitory and excitatory synaptic transmission and can contribute to cell proliferation, memory storage, and/or access [65]. Additionally, BDNF has been implicated as a key regulator in the memory reconsolidation pathway for modifying remote memory [66]. CREB1 is a crucial transcription factor in remote memory formation [67,68]. A study on CREB functionality in medial prefrontal cortex cells demonstrated CREB-dependent transcription in relation to remote memory [69]. MAPK4 is a recently identified mitogen-activated protein kinase linked to remote memory formation [70,71,72].

We utilized the expression profiles of FC-pos cells and FC-neg cells for GRN inference, resulting in the prediction of 5, 6, and 1 regulations for BDNF, CREB1, and MAPK4, respectively. To verify the edges for the first neighbors of the genes of interest, we conducted a literature search. Out of the 12 significant interactions detected, 8 were supported by the literature (Figure 4, Appendix A). Notably, all 4 edges in the GRN of BDNF were validated by literature evidence (Figure 4a, Appendix A). When scTIGER was applied to CREB1, 6 significant interactions were detected, and 4 of them were supported by the literature (Figure 4b, Appendix A). Only one significant interaction was detected for MAPK4 (Figure 4c, Appendix A). Overall, the small numbers of predicted regulations and high proportions of literature supported results demonstrate that scTIGER can accurately identify true edges with directionality and potentially discover new edges, while minimizing false positives for genes that lack cell- or condition-specific functions.

### 2.6. Robustness against Dropouts and Cell Orders

Dropout counts, which occur when a gene is expressed in one cell but not in another cell of the same type, are a consequence of the low capture efficiency inherent in scRNA-seq methods. To assess the reliability of scTIGER in inferring GRNs, we conducted experiments using artificially inserted dropouts at various levels in real scRNA-seq data. In the original scRNA-seq dataset, the dropout level was already substantial, reaching approximately 90%. By introducing 1% artificial dropouts, we examined whether scTIGER could consistently generate reliable GRN predictions. Remarkably, the GRN associated with the AR exhibited a highly conserved pattern, with a median overlap of 100% and a minimum overlap of 83.33% across 10 independent runs (Appendix A). When the dropout level was increased to 2%, the median overlap decreased to 66.67%. However, when 5% artificial dropouts were inserted, the overlap dropped to 25%. These findings suggest that scTIGER demonstrates remarkable robustness when the dropout levels remain below 91%, which is typically the upper limit for most current scRNA-seq datasets.

We then tested scTIGER to determine if there was a significant difference when calculating the differential expressions by using different orders of two datasets (e.g., case–control or control–case). For example, scTIGER was implemented by using ECs as the case condition and SMCs as the control condition (Figure 3). When they were flipped, the detected GRNs contain mostly the same interactions (Appendix A).

To ensure the reliability of our pseudotiming approach in ordering cells, it is important to investigate whether different cell orders initially inputted would lead to different predictions. To test this, we randomly changed the column positions (i.e., cells) and reran scTIGER. However, no changes in the results were observed, indicating that the pseudotiming procedure can accurately reconstruct the cell orders. As a benchmarking measure, we also performed a random shuffling of the input gene expression matrices and reran scTIGER. However, we failed to detect any significant regulations. These validation tests collectively demonstrate that scTIGER has the ability to infer potential regulations that would otherwise remain hidden within scRNA-seq data.

## 3. Discussion

We developed a novel method called scTIGER and demonstrated its effectiveness in uncovering regulatory information from scRNA-seq data. By leveraging pseudotiming to order cells, scTIGER reliably reconstructed cell orders, ensuring accurate predictions of regulatory interactions. Applying scTIGER to diverse datasets enabled us to investigate common potential regulations among different cell types within the same disease context. By focusing on specific genes of interest, such as AR, PTEN, and ERG, known to impact prostate cancer progression and metastasis, we discovered novel regulatory interactions. Importantly, scTIGER successfully identified known interactions reported in the literature, validating our approach. We extended the application of scTIGER to investigate neural and neural-associated cells involved in fear conditioning. By selecting BDNF, CREB1, and MAPK4, known to play roles in memory formation, consolidation, and recall, we uncovered regulatory interactions related to remote memory processes. The literature verification of these detected interactions confirmed the reliability of scTIGER in capturing biologically meaningful regulatory relationships. Notably, scTIGER’s ability to reveal previously unknown edges suggests its potential in discovering novel regulatory connections that may have been overlooked in traditional analyses.

An intriguing observation is the identification of commonly regulated genes across different GRNs. For instance, the gene SH3BGRL2 is found to be regulated by both PTEN and AR (Appendix A). Additionally, we observe that multiple genes, PDCD7, CFI, and KIF1B, are commonly regulated by both AR and ATF3 (Appendix A). These findings highlight potential connections between different GRNs, indicating a coupled regulatory influence on specific genes. Such information is invaluable for further investigations into co-regulation mechanisms, offering insights into potential combinatorial drug targets. By understanding the shared regulatory patterns of these genes, we can explore targeted therapeutic approaches that address multiple pathways simultaneously. This integrative approach has the potential to enhance treatment efficacy and develop more precise interventions in the field of molecular medicine.

Although the real applications on scRNA-seq of prostate cancer and engram neurons are quite promising, one limitation of this research is the absence of benchmarking datasets for case-control studies. To address this, we propose generating simulation datasets as a valuable approach to validate and benchmark the performance of scTIGER in case-control studies. Simulated datasets allow us to have control over the underlying regulatory relationships, noise levels, and other factors, providing a baseline for comparison and evaluation. In brief, we can define the regulatory relationships among target genes and their corresponding regulators, simulate gene expression data based on these relationships and experimental conditions using mathematical models [29,73] such as differential equations or Boolean networks, and introduce appropriate levels of noise to mimic the variability and measurement errors observed in scRNA-seq data. Additionally, we can incorporate technical variations typical in scRNA-seq experiments, such as dropout events and amplification biases, to make the simulated datasets more realistic. With these simulation datasets, we can apply scTIGER and assess its performance in inferring the underlying regulatory relationships. By comparing the predicted regulatory interactions with the known interactions in the simulated data and evaluating various performance metrics such as precision, recall, and F1 score, we can quantify the accuracy of scTIGER’s predictions. We plan to conduct such simulation studies in the near future.

In conclusion, scTIGER offers the first tool for deciphering hidden regulatory information from case versus control scRNA-seq data. Its ability to accurately reconstruct cell orders, detect specific regulatory interactions, and avoid false positives highlights its effectiveness in inferring potential gene regulatory networks. By providing insights into cell-specific regulations and uncovering novel regulatory patterns, scTIGER contributes to our understanding of disease mechanisms and offers opportunities for the identification of new therapeutic targets and treatment strategies.

## 4. Materials and Methods

### 4.1. scTIGER Overview and Design

scTIGER is designed to construct GRNs using paired scRNA-seq datasets that are widely available in diverse biological research. It mainly combines cell-type-based pseudotiming [24], temporal causal discovery framework (TCDF) [25], and gene-specific permutation-based significance testing to infer causal regulatory relationships among genes (Figure 1a). Firstly, scTIGER takes two scRNA-seq datasets as input, such as case and control data. It initially normalizes the data and clusters the cells by using the Leiden algorithm in SCANPY (version 1.9.1). For a given cell type, scTIGER performs pseudotiming on their cells to obtain the chronological cell orders, and then calculates the differential expression levels of genes. If one dataset has more cells than another, its cells are randomly selected to match the cell numbers in the other dataset. Secondly, scTIGER employs the TCDF method to predict causality by using the differential expression vectors of given gene sets. TCDF uses attention-based convolutional neural networks combined with a causal validation step, and it has been evaluated to be highly accurate and reliable [25]. When examining a particular gene, scTIGER searches for the top k genes with highly correlated differential expression vectors. The parameter k is user-selectable and can enhance both the search performance and running speed, particularly when a smaller score is defined (e.g., 100 used in this research). If a significant causality between a gene pair is detected, scTIGER outputs the causality and number of delayed steps between them. Thirdly, positive regulation between two genes is defined as a positive correlation of their differential expression vectors, while negative regulation is defined as a negative correlation. To further remove false-positive predictions and perform significance testing, we repeated random sampling of cells multiple times and reran the prediction procedure. A statistical model (negative binomial) is then constructed to estimate the frequency of how many times a prediction can be recalled in 100 repeats.

Unlike traditional GRN inference methods that primarily use co-expression information, scTIGER uses the gene differential expression matrix to filter out false positives. For example, two genes may have a high correlation in one experiment but have no co-differential changes under two experiments (e.g., the $g_{j}$ and $g_{k}$ in Figure 1b). Traditional methods may predict that they are causally related. However, scTIGER avoids this by noting that the differential expression levels of $g_{j}$ and $g_{k}$ have no clear patterns and are likely to be white noise. Therefore, they will have no causalities with other genes that have clear dynamical patterns (e.g., $g_{i}$, $g_{s},$ and $g_{t}$, the right side in Figure 1b). This strategy can not only filter more false positives, but also provide condition-specific or cell-specific regulations. It is important to apply in case vs. control studies to understand the novel mechanisms underlying cancer development. For a given gene set, scTIGER outputs the regulated genes for each gene, as well as a merged network by combining the results of all genes. It can be used to predict condition-specific GRNs for the same cell types and cell-specific GRNs across different cell types within the same condition (Figure 1c). This is especially useful to understand how different cell types exhibit similar mechanisms in one specific condition, such as the health and cancer microenvironment, and drug treatment.

### 4.2. Data Preparation and Pseudotiming

scTIGER takes two scRNA-seq datasets as input, typically case and control data. The first step involves data normalization and the identification of low-quality cells and genes, following similar strategies as SCANPY [74]. Specifically, cells with low gene expression counts or low total counts are filtered out. Additionally, cells exhibiting high mitochondrial gene expression, which may indicate poor-quality cells, are excluded. Genes with low expression counts across cells are also filtered out to reduce noise. Next, scTIGER performs pseudotiming on the normalized cells to determine their chronological cell orders using PAGA [24], utilizing the built-in function "scanpy.tl.paga" in SCANPY). In cases where the two datasets have unequal numbers of cells, scTIGER employs a random selection approach during each permutation to ensure an equal number of cells for calculating paired differential expression levels of genes.

### 4.3. Detecting Causal Interactions

When examining a particular gene or genes, scTIGER searches for the top genes with highly correlated differential expression vectors by calculating their Pearson correlations. The number of top genes is user-selectable and can enhance both the search performance and running speed, particularly when a smaller score is defined (e.g., top 100 genes in this study). scTIGER utilizes the TCDF method [25] to predict causality by using the differential expression vectors of selected top genes that are highly correlated to the gene to be analyzed. TCDF uses attention-based convolutional neural networks combined with causal validation steps, and has demonstrated high accuracy and reliability in temporal causal discovery [25]. In the context of scTIGER, the dataset X comprises N observed time series (i.e., differential expression profiles of N genes) of the same length T (i.e., $X=\left(X_{1},X_{2},\ldots ,X_{N}\right)\in R^{N\times T}$), where each $X_{i}=\Delta e\left(g_{i}\right)=e_{case}\left(g_{i}\right)-e_{control}(g_{i})$. The objective of temporal causal discovery is to identify causal relationships between all N time series in X and determine the time delay between cause and effect. The predicted causal relationships can be represented as a temporal causal graph G=(V,E), where $v_{i}\in V$ represents a gene and each directed edge $e_{i,j}\in E$ from gene $v_{i}$ to $v_{j}$ denotes a causal relationship from $v_{i}$ to $v_{j}$.

The networks used in TCDF are called attention-based dilated depthwise separable temporal convolutional networks (AD-DSTCNs) [25]. More specifically, TCDF consists of N separate convolutional neural networks (CNNs), each for a time series $X_{j}$. The N CNNs can be regarded as N independent and structurally identical temporal convolutional networks (TCNs). One CNN is only trained to predict one time-series profile, based on the past values of all time-series profiles. The internal parameters of each CNN are trained by backpropagation and then used for unsupervised causal discovery. The jth TCN corresponds to the target time series $X_{j}$ and all other TCNs correspond to the exogenous time series $X_{i\neq j}$; all time-series of length T should be inputs; the outputs are the predicted ${\hat{X}}_{j}$ of $X_{j}$, the kernel weight $W_{j}$ and attention scores $a_{j}$, and the loss function is the mean squared error (MSE) loss between $X_{j}$ and $\hat{X_{j}}$. The attention score vector $a_{j}=[a_{1j},a_{2j},\ldots ,a_{Nj}]$ of the jth TCN is element-wise multiplied with the N input time series, where $a_{ij}$ represents how much the jth network intends to input of $X_{i}$ for predicting target $X_{j}$. A TCN predicts each time step of the target time series $X_{j}$ by sliding a kernel over an input $X_{i}=[x_{i}^{1},x_{i}^{2},\ldots ,x_{i}^{T}]$. When predicting the value of $X_{j}$ at time step t (i.e., $x_{j}^{t}$), the kernel with a kernel size K calculates $W⊙\left[x_{i}^{t-K+1},x_{i}^{t-K+2},\ldots ,x_{i}^{t-1},x_{i}^{t}\right]$ if i≠j, or $W⊙\left[x_{i}^{t-K},x_{i}^{t-K+1},\ldots ,x_{i}^{t-1}\right]$ if i=j, where W is the learnt kernel weights and $x_{i}^{l}=0$ if l≤0. PReLU is used as a non-linear activation function in each hidden layer of a TCN. Each attention score $a\in a_{j}$ is truncated to zero if a is less than a threshold $\tau_{j}$ and applied by Softmax function if $a\geq \tau_{j}$. Thus, a set of potential causes for each time series can be created after comparing all attention scores. An architecture example of deep-learning approaches in scTIGER is shown in Appendix A.

Furthermore, the convolution layers of the TCNs are dilated convolutions. The kernel of a dilated convolution should be applied to an area larger than the kernel’s size by skipping a step size f, where $f=c^{l}$ for layer l and c is the dilation coefficient. In addition, a residual connection is added after each convolution layer excluding the first layer in each TCN. Permutation importance (PI) is used as a causal validation method, which is based on the assumption that the predictions should be worse if the values of a true cause are randomly permuted. The ΔL is the difference between the initial loss at the first epoch and the loss when the trained network is applied to the permuted dataset. If the $\Delta L\leq 0.8L_{G}$ where $L_{G}$ is the ground loss, then we conclude the current potential cause is a true cause. The pseudotime step delay between cause $X_{i}$ and target $X_{j}$ equals the position of the highest kernel weight. Note that we train all AD-DSTCNs by using the Adam optimizer for 5000 epochs, with a learning rate $10^{-2}$, kernel size K=4, dilation coefficient c=4, and the number of hidden layers L=2.

If a significant causal relationship is detected between a pair of genes, TCDF reports the causality along with the number of delayed steps between them. scTIGER utilizes this information to selectively identify direct causal relationships between genes. By default, scTIGER focuses on predicting direct causal interactions, but it can be adjusted to include indirect interactions as well. Exploring indirect interactions can offer valuable insights into potential network expansions, thereby enhancing discovery capabilities. The regulatory relationship between two genes is further classified as either positive or negative regulation. Positive regulation is defined as a positive correlation between their differential expression vectors, while negative regulation is characterized by a negative correlation.

### 4.4. Permutation Testing

To further mitigate false positive predictions and conduct significance testing, we employed repeated random sampling of cells and re-executed the procedure. The frequency at which a prediction is consistently recalled in a user-defined number of repetitions is compared to the frequency of a recalled prediction in a background run. scTIGER incorporates permutation testing, which inherently includes bootstrapping, enabling automatic assessment of confidence in each edge. The number of identified edges at different confidence levels, determined through permutation testing, is subjected to a negative binomial model to calculate the minimum number of detections required for an interaction to be deemed significant. For a given gene set, scTIGER provides the GRNs for each individual gene, as well as a merged GRN obtained by combining the results of all genes after removing background interactions.

### 4.5. Benchmarking against Existing Methods

We utilized synthetic and curated datasets which provide a list of true positive interactions (available at https://doi.org/10.5281/zenodo.3378975, accessed on 31 July 2023). These datasets were previously used to compare many GRN models such as PIDC, SINCERITIES, SCODE, GRNBoost2, and PPCOR among others. We used two synthetic datasets mimicking a linear trajectory (LI) and cyclical trajectory (CY), and two curated datasets for hematopoietic stem cell differentiation (HSC) and mammalian cortical area development (mCAD). All datasets contained 200 cells to perform the benchmarking. We assess scTIGER’s performance in relation to the existing models PIDC, SINCERITIES, SCODE, GRNBoost2, and PPCOR using these datasets.

Due to the lack of benchmarking datasets for paired case versus control data, the pipeline was modified by removing the differential expression matrix calculation and correlation steps. Thus, the benchmarking only covers the inference of edges using the deep-learning algorithm, permutation, and significance threshold calculation steps in scTIGER. Meanwhile, assessing edge recovery, edge directionality, sign, and self-loops were disregarded to measure performance in an identical manner for all methods. To compare the performances of these six methods, we calculated the precision, recall, F1 score, and specificity for each on all datasets.

### 4.6. Validations on Real scRNA-seq Datasets

To validate the performance of scTIGER in detecting experimentally validated regulations, we conducted two real case studies involving prostate cancer and neurons with remote memory. In the first case study, we utilized scTIGER on scRNA-seq datasets (GSE193337) obtained from patient samples with prostate cancer [30]. Initially, we identified potential regulations by using the co-differential expression profiles between prostate cancer samples and corresponding normal prostate cells. Subsequently, we focused on detecting common regulations between tumorigenic endothelial cells (ECs) and smooth muscle cells (SMCs). Cell types were annotated using the ScType [75] and for downstream analysis, we selected 97 ECs and 96 SMCs. To match the EC count in each permutation, 97 cells were randomly sampled from the corresponding normal tissue sample, which contained a total of 447 cells. In the second case study, we applied scTIGER to scRNA-seq datasets (GSE152632) obtained from mouse neurons in the medial prefrontal cortex, comparing neurons with and without fear memories [76]. Specifically, we examined regulatory mechanisms related to memory formation by using the expression profiles of neurons subjected to fear training using shocking (FC-pos) and a control group where mice underwent cage transfer without any shocking (FC-neg). For each permutation, we randomly selected 192 FC-neg cells from a pool of 329 cells and an equal number of FC-pos cells.

### 4.7. Testing Robustness against Dropout Noise

To evaluate the robustness of scTIGER against dropout noises, we introduced artificial modifications to the prostate cancer scRNA-seq dataset by incorporating varying levels of dropout noise. This was achieved by replacing a certain number of counts with artificial zeros, corresponding to a predetermined proportion of dropout counts. It is important to note that the original datasets already contained inherent levels of dropouts. For each paired dataset, we generated independent datasets with different artificial dropout proportions, gradually increasing the total dropout proportions up to 0.95. Subsequently, we assessed the overlap of predicted regulations for a targeted gene across all predictions, considering different dropout levels as well as the original dataset.

## Figures and Tables

**Figure 1 ijms-24-13339-f001:**
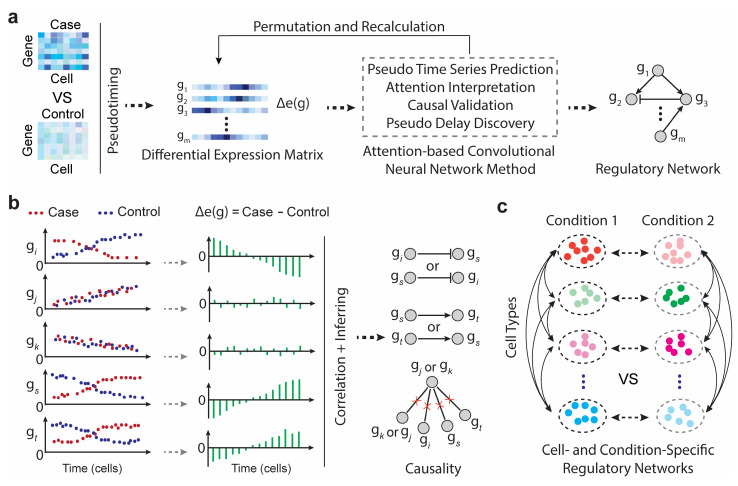
Computational workflow and applications of scTIGER. (**a**) The main workflow is depicted. The predicted causal relationships among genes are presented as a directed causal graph (i.e., regulatory network). (**b**) Correlations in co-differential expression among genes can be used to determine causal relationships. Genes that exhibit correlated expression patterns but lack co-differential correlations between the two conditions are theoretically excluded from the regulatory network. (**c**) scTIGER has the capability to detect regulatory networks by comparing different cell types and conditions. Distinct colors represent different cell types, while the same color with either a dark or light style signifies identical cell types in two conditions.

**Figure 2 ijms-24-13339-f002:**
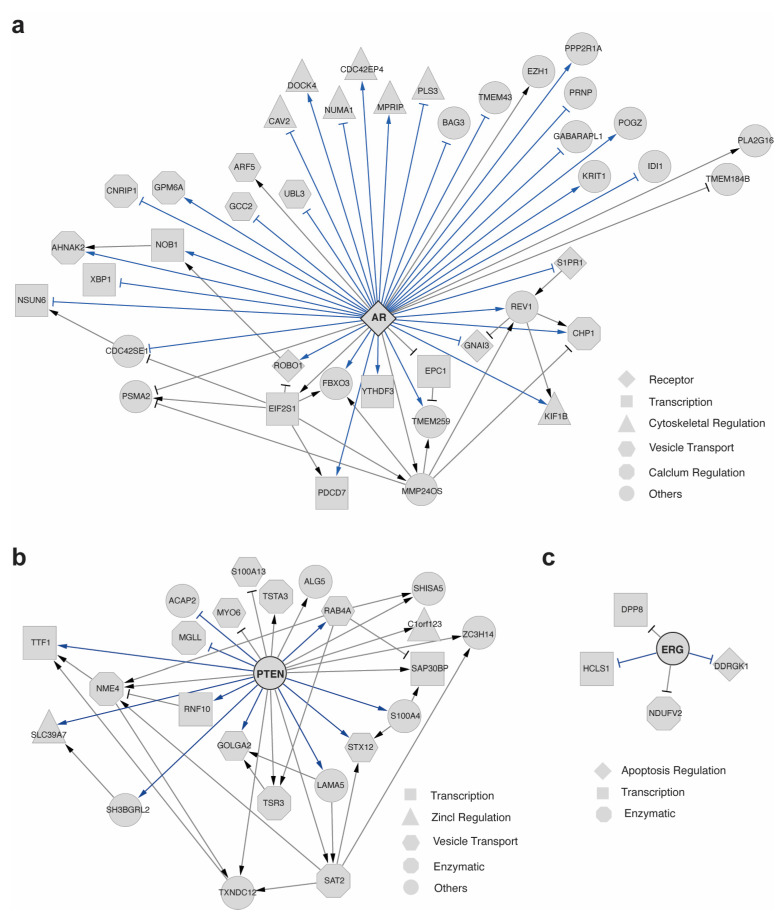
Predicted GRNs of AR, PTEN, and ERG in ECs. GRNs show the regulatory relationships by comparing cancerous versus normal conditions for (**a**) AR, (**b**) PTEN, and (**c**) ERG. Shapes represent the processes in which each gene’s corresponding gene product is involved. Only nodes directly connected to the genes of interest are displayed. The arrowheads and T-heads indicate the positive and negative regulation, respectively. Blue edges denote literature verified relationships and grey edges denote potential novel regulations.

**Figure 3 ijms-24-13339-f003:**
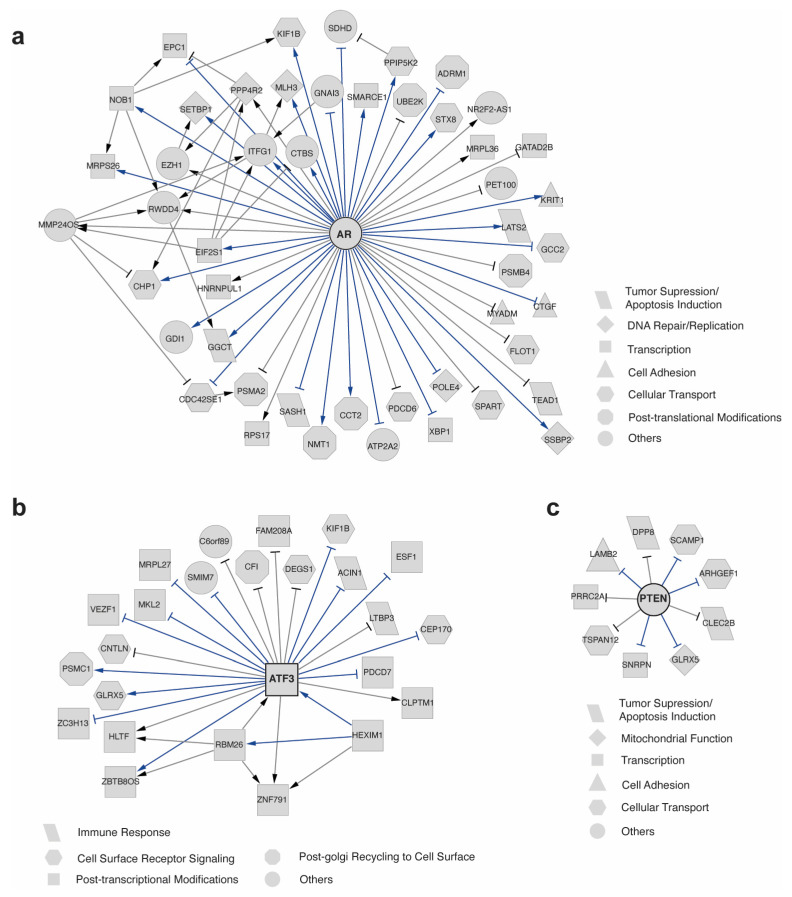
Predicted GRNs of AR, PTEN, and ATF3. GRNs show the regulatory relationships by comparing ECs and SMCs under cancerous condition for (**a**) AR, (**b**) ATF3, and (**c**) PTEN. Shapes represent the processes in which each gene’s corresponding gene product is involved. Only nodes directly connected to the genes of interest are displayed. The arrowheads and T-heads indicate the positive and negative regulation, respectively. Blue edges denote literature-verified relationships and grey edges denote potential novel regulations.

**Figure 4 ijms-24-13339-f004:**
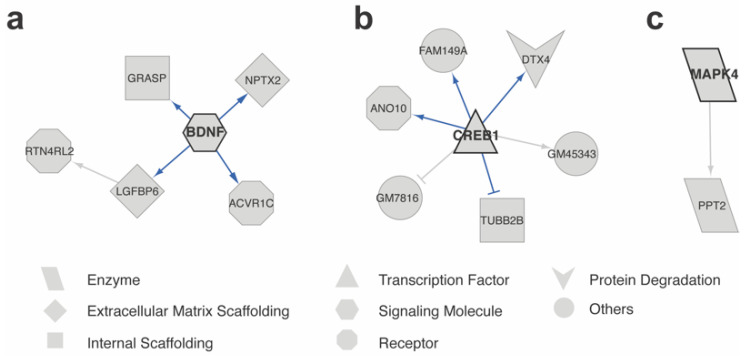
Predicted GRNs of BDNF, CREB1, and MAPK4. GRNs show the regulatory relationships potentially involved in remote memory formation by comparing FC-pos and FC-neg neurons for (**a**) BDNF, (**b**) CREB1, and (**c**) MAPK4. Shapes represent the processes in which each gene’s corresponding gene product is involved. Only nodes directly connected to the genes of interest are displayed. The arrowheads and T-heads indicate the positive and negative regulation, respectively. Blue edges denote literature-verified relationships and grey edges denote potential novel regulations.

**Table 1 ijms-24-13339-t001:** scTIGER properties compared with existing models.

Method	Category	Directed	Signed	Additional Required Input
scTIGER	Corr	✓	✓	-
PIDC	MI	✘	✘	-
SINCERITIES	Reg	✓	✓	-
GRNBoost2	RF	✓	✘	-
PPCOR	Corr	✘	✓	-
SCODE	ODE + Reg	✓	✓	ODE Parameters

Note. MI: Mutual Information, RF: Random Forest, Corr: Correlation, ODE: Ordinary Differential Equations, Reg: Regression. “-” indicates not required.

**Table 2 ijms-24-13339-t002:** Comparing scTIGER performance with five existing models.

Method	Criteria	LI	CY	HSC	mCAD
scTIGER	Recall	1	1	0.667	0.667
	Precision	1	0.75	0.56	0.85
	Specificity	1	0.778	0.676	0
	F1 Score	1	0.857	0.609	0.75
PIDC	Recall	1	1	0.619	0.667
	Precision	0.583	0.75	0.52	0.85
	Specificity	0.643	0.778	0.647	0
	F1 Score	0.636	0.857	0.565	0.75
SINCERITIES	Recall	0.286	0.667	0.667	1
	Precision	0.25	0.4	0.452	0.9
	Specificity	0.571	0.333	0.5	0
	F1 Score	0.267	0.5	0.538	0.947
GRNBoost2	Recall	0.857	1	0.667	1
	Precision	0.667	0.75	0.56	0.9
	Specificity	0.786	0.778	0.676	0
	F1 Score	0.75	0.857	0.609	0.947
PPCOR	Recall	0.857	0.667	0.190	0.444
	Precision	1	0.667	0.571	0.8
	Specificity	1	0.778	0.912	0
	F1 Score	0.923	0.667	0.286	0.571
SCODE	Recall	0.571	0.333	0.667	1
	Precision	0.5	0.25	0.452	0.9
	Specificity	0.714	0.333	0.5	0
	F1 Score	0.533	0.286	0.538	0.947

## Data Availability

Data used in this study are available in NICBI GEO with the accession number GSE193337 for prostatic cells and GSE152632 for neurons with and without remote memory. The synthetic and curated datasets are available in https://doi.org/10.5281/zenodo.3378975 (accessed on 31 July 2023). The source code and usage instructions of scTIGER are publicly available at https://github.com/chenyongrowan/scTIGER (accessed on 31 July 2023).

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
