# Peer review of "scTIGER: A Deep-Learning Method for Inferring Gene Regulatory Networks from Case versus Control scRNA-seq Datasets"

_ijms, 2023, doi:10.3390/ijms241713339_

Round 1
Reviewer 1 Report
In this manuscript, the author has developed a novel method, scTIGER, for inferring gene regulatory networks from single cell RNA-seq data. The author used this method to detect various gene networks from single cell RNA-seq datasets of prostate cancer cells and neurons.
One of the features of scTIGER described in this manuscript is that it can minimize false positives and false negatives compared to other methods such as WGCNA, SCENIC and SCODE. Specifically, it would be good to indicate the degree of improvement compared to other softwares.
Reviewer 2 Report
The authors present scTIGER, a GRN inference algorithm for case versus control single cell data. This tool employs a deep learning framework and solves issues of false positive results by taking into account differentially expressed genes as well as pseudotime inference. They provide an excellent introduction and motivation for their study, however, both the methods as well as results presented are lacking.
Major comments:
1. Fig 1 is presented at the very end with the methods, and Fig1a is not mentioned anywhere in the text. This is highly atypical in a biological manuscript
2. This is a methods paper and the authors do very little to describe what deep learning approach and architecture they use. The acronym TCDF is mentioned in one section and explained in another section but there is no evaluation of performance of the model itself compared to other similar models. I also did not see a description of the training and testing set used for this framework that makes it highly suspicious
3. There is no benchmarking done against various other tools that perform GRN inference
4. The authors do not provide any description of how the examples they present in their figures were chosen
Minor comments:
1. There is no description of the acronym scTIGER
2. The authors should pay attention to grammar in some parts
